# The Development of Hindlimb Postural Asymmetry Induced by Focal Traumatic Brain Injury Is Not Related to Serotonin 2A/C Receptor Expression in the Spinal Cord

**DOI:** 10.3390/ijms23105358

**Published:** 2022-05-11

**Authors:** Marlene Storm Andersen, Dilârâ Bedriye Güler, Jonas Larsen, Karen Kalhøj Rich, Åsa Fex Svenningsen, Mengliang Zhang

**Affiliations:** 1Department of Molecular Medicine, University of Southern Denmark, DK-5000 Odense, Denmark; m_storm_a@hotmail.com (M.S.A.); dilara@guler.dk (D.B.G.); jonaslarsen2008@hotmail.com (J.L.); krich@health.sdu.dk (K.K.R.); aasvenningsen@health.sdu.dk (Å.F.S.); 2BRIDGE, University of Southern Denmark, DK-5000 Odense, Denmark

**Keywords:** traumatic brain injury, postural asymmetry, gait deficit, serotonin, serotonin receptor

## Abstract

Brain injury and stroke are leading causes of adult disability. Motor deficits are common problems, and their underlying pathological mechanisms remain poorly understood. The serotoninergic system is implicated in both functional recovery from and the occurrence of spasticity after injuries to the central nervous system. This study, which was conducted on rats, investigated the development of limb postural changes and their relationship to the expression of serotonin (5-HT) 2A and 2C receptors in the spinal cord in the 4 weeks after focal traumatic brain injury (TBI) to the right hindlimb sensorimotor cortex. The limb motor deficits were assessed by measuring gait pattern changes during walking and hindlimb postural asymmetry at different time intervals (3–28 days) after surgery. The expressions of the 5-HT2A and 2C receptors in the lumbar spinal cord were investigated using immunohistochemistry. The results showed that all the rats with TBI, independently of the duration of the interval, displayed postural asymmetry with flexion on the contralateral (left) side (>2 mm), while the sham-operated rats showed no apparent postural asymmetry. The TBI rats also had longer stride lengths during walking in both their hindlimbs and their forelimbs compared with the sham rats. For both the TBI and the sham rats, the hind-paw placement angles were larger on the contralateral side in some of the groups. Compared to the sham-operated rats, the 5-HT2A and 2C receptor expression did not significantly change on either side of the lumbar spinal cords of the TBI rats in any of the groups. These results suggest that focal TBI can induce motor deficits lasting a relatively long time, and that these deficits are not related to the expression of the 5-HT2A and 2C receptors in the spinal cord.

## 1. Introduction

Traumatic brain injury (TBI) and stroke are leading causes of adult disability, often manifesting as spasticity that leads to gait and postural changes and prevents patients from returning to their previous level of functioning [1,2,3,4]. The pathological mechanisms underlying these motor deficits are not fully understood, and there is no effective treatment to facilitate functional recovery from and prevent the progression of physical disability. Identifying the neurobiological mechanisms underlying these motor deficits may be the key to the development of new therapeutics to promote motor function recovery through, for example, the manipulation of neural plasticity [5].

TBI and stroke mostly occur unilaterally and often result in postural asymmetry (PA) [6,7,8]. One of the underlying mechanisms of PA may be the asymmetric activity of descending neural tracts and subsequent plastic rearrangements of spinal neurocircuits [9,10,11,12]. For example, unilateral TBI-induced PA persists after complete spinal transection, demonstrating the critical role of spinal plasticity in the persistence of such symptoms [6,7,8]. Spinal plasticity also occurs after spinal cord injury, and monosynaptic reflexes are enhanced on the ipsilateral side after spinal cord hemisection; these reflex changes remain after the transection of the spinal cord at higher levels [13,14,15,16,17].

In addition to changes in spinal neurocircuits, it has been suggested that plastic changes in monoamines and their receptors may be among the factors that cause motor deficits. Serotonin (5-HT) receptors, especially 5-HT2A and 5-HT2C receptors (5-HT2AR, 5-HT2CR), appear to be among the most important receptors in the spinal cord with respect to the regulation of normal motor functions and functional recovery after spinal cord injury [18,19,20,21,22,23]. Exaggerated motor activity can be caused by the supersensitivity of 5-HT receptors [24,25] and after complete spinal cord injury, 5-HT2AR and 5-HT2CR undergo a long-lasting upregulation that may play a role in the generation of spasticity [26,27,28,29]. While many studies have investigated the changes in the expression of 5-HT receptors in relation to spinal cord injury, we know of no study that has examined their expression after brain injury. Although some changes are similar after spinal cord injury and brain injury (e.g., the $K^{+}/{\mathrm{Cl}}^{-}$ cotransporter KCC2 has been found to be downregulated after both kinds of injuries [30,31]), the two injury types might result in different plastic changes in the spinal cord and, thus, have different effects on expression of 5-HT receptors.

In this study, we used an established unilateral TBI animal model in which the hindlimb representation area of the sensorimotor cortex was ablated [8]. Although this TBI model has been shown to induce hindlimb postural asymmetry (HL-PA) on the contralateral side for varying periods after injury, a time course study of its development is lacking [8]. We further examined whether HL-PA changed over a period of 4 weeks and whether the injury affected the animals’ walking patterns over the same period. Finally, we examined whether the expression of 5-HT2AR and 5-HT2CR in the lumbar spinal cords of the rats was affected, with the aim of evaluating their potential role in the development of motor deficits after TBI.

## 2. Results

### 2.1. Time Course of Hindlimb Postural Asymmetry (HL-PA) Development over 4 Weeks

As seen in Figure 1, the HL-PA in the TBI rats showed a contralateral flexion of the hindlimb that was of significantly larger amplitude compared to the sham-operated rats. Despite individual variations, the average HL-PA amplitude was over 2 mm for the TBI rats in all the groups (3 days 2.2 ± 2.1; 7 days 3.6 ± 2.1 mm; 14 days 2.7 ± 1.5 mm; 21 days 4.7 ± 0.5 mm; and 28 days 3. 7 ± 1.2 mm), whereas the average HL-PA amplitude was less than 1 mm in the sham rats in all the groups, except at 14 days (1.1 ± 0.7 mm). However, these differences in HL-PA amplitude between the TBI rats and the sham rats were only significant at 7, 21 and 28 days, but not at 3 or 14 days. The largest HL-PA was seen in the TBI rats at 21 days (Figure 1B).

To remove the effects of covariates from both the TBI and the sham group, a difference in HL-PA between TBI and sham was presented in each time group. As seen in Figure 1C, the differences in HL-PA amplitude were still more than 2 mm at 7, 21, and 28-days (2.7 ± 1.7, 4.3 ± 2.4, and 2.8 ± 1.8 mm, respectively), whereas they were slightly less than 2 mm at 3 days and 14 days (1.8 ± 1.7 and 1.6 ± 1.3 mm). We saw no clear trend in the time course of the development of asymmetry. Significant differences in HL-PA were only seen when comparing 3 days versus 21 days, and 14 days versus 21 days. The HL-PA thus did not develop further, but remained at a significantly higher level even 4 weeks after the injury.

### 2.2. Gait Pattern after Traumatic Brain Injury (TBI) and Sham Surgery

#### 2.2.1. TBI Rats Had a Longer Stride Length Than Sham Rats

In the TBI and sham rats surviving 28 days, we analyzed the stride lengths (SLs) of both the forelimbs and the hindlimbs at different time points over the 28 days (Figure 2, see also Figure 10). At 3 days after injury, the forelimb and hindlimb SLs were generally longer for the TBI rats than for the sham rats for both the ipsilateral and the contralateral side, and this tendency remained until 28 days. However, there were no significant differences in forelimb or hindlimb SL between the ipsilateral and contralateral sides at any time point for either the TBI or the sham rats (Figure 2). Thus, we decided to average the values from the two sides to compare the SLs between the TBI and sham rats. As shown in Figure 2, the SL stayed relatively stable in the sham rats, but it increased in the TBI rats for both the forelimb and the hindlimb. The forelimb SLs were approximately 1.10–1.24-fold longer in the TBI rats than in the sham rats at 3 to 28 days, and the difference was significant at 7, 21 and 28 days. The hindlimb SLs were 1.11–1.31-fold longer in the TBI rats than in the sham rats at 3 to 28 days, and the difference was significant at 3, 14 and 28 days (Table 1).

#### 2.2.2. Both TBI and Sham Surgery Changed the Hind-Paw Placement Angle

The hind-paw placement angle was analyzed on the ipsilateral and contralateral sides at different time points in the TBI and sham rats surviving 28 days (Figure 3, see also Figure 10). The paw placement angle on the contralateral side was larger than on the ipsilateral side in both the TBI and the sham rats, but this difference was not always significant (Figure 3A,B). For the TBI rats, a significant difference was only seen at 28 days post-injury, when the contralateral hind paw was 1.4-fold more angled (21 degrees) than the ipsilateral hind paw (15 degrees) (Figure 3A,C). For the sham-operated rats, the contralateral hind paw was significantly more angled (1.50-, 1.53-. and 2.00-fold larger) than the ipsilateral hind paw after 7, 21, and 28 days (i.e., 18, 20, and 22 degrees vs. 12, 13, and 11 degrees) (Figure 3B,D).

When examining the time course of the hind-paw placement angle changes, the TBI rats showed a sustained increase in hind-paw angle on both the ipsilateral and the contralateral side. Thus, compared with the angle before injury (0 days), the contralateral side had a significantly larger angle (1.4–1.9-fold) at all time intervals (Figure 3E), while the ipsilateral hind paw had a significantly larger angle (approximately 1.4–1.7-fold) at all time intervals except for 14 days (Figure 3G). In the sham rats, a significant difference was only seen on the contralateral side and not the ipsilateral side (Figure 3F,H). Thus, the contralateral hind paw was significantly more angled at 7, 21, and 28 days (1.4-, 1.5-, and 1.7-fold) (Figure 3F).

### 2.3. No Significant Changes in 5-HT2AR/2CR Expression in Lumbar Spinal Cord after TBI

Consistent with previous reports [27,29], both 5-HT2AR and 2CR were expressed in the lumbar spinal cords of the rats. The 5-HT2ARs were mainly expressed in the ventral horn motoneuron region, although less intense expression was also detected in other regions in the gray matter. The 5-HT2CRs were widely expressed in different regions of the gray matter. As our study focused on the ventral horn motoneurons, we confirmed 5-HT2AR/2CR expression on the motoneurons by double immunostaining with choline acetyltransferase (ChAT) antibody, which is a motoneuronal marker. As shown in Figure 4, 5-HT2AR and 2CR expression was seen in the ventral horn motoneurons.

One of the main purposes of the present study was to determine whether focal brain injury would alter 5-HT2AR and 2CR expression in the spinal cords of the rats. We achieved this by analyzing the 5-HT2AR/2CR immunoreactive density through white–black images. As illustrated in Figure 5 and Figure 6, equal-sized areas from the corresponding regions of the ventral horn were selected from both sides of the TBI and sham rats, and the optical density was measured with ImageJ. First, we examined whether TBI and sham surgery would cause differential 5-HT2AR/2CR expression on the contralateral and ipsilateral sides. To this end, the optical density on the contralateral side was divided/normalized by the density on the ipsilateral side. The results showed that in both the TBI and the sham rats, the ratio of the optical density was close to 1 for both 5-HT2AR and 5-HT2CR. There were no significant differences between the two sides in either the TBI or the sham groups (Figure 7A,B).

Next, we examined whether the 5-HT2AR/2CR expression differed between the TBI and the sham rats on the same side of the spinal cord. To this end, the optical density in the TBI rats was divided by that in the sham rats on the same side. The results showed that for both 5-HT2AR and 2CR, there were no significant differences between the TBI and sham rats on either side in any of the time groups. As shown in Figure 7C,D, there were no clear trends in the expression of the two receptors, although some fluctuations were seen. For example, 5-HT2AR expression showed a clear decrease in the 7-day TBI rats on both sides compared with the 3-day TBI rats (ca. 79%). This led to a significant difference between day 7 and all other the days for both sides. From 14 days, 5-HT2AR expression recovered to close to sham level and remained at that level. The 5-HT2CR showed a lower density in the 3-day TBI rats compared with the sham rats (ca. 80%) for both sides, but an increased density in the 7-day TBI rats compared with the 3-day TBI rats (ca. 140%), which was slightly over the sham level. This increase led to a significant difference between the 3-day and 7-day TBI rats for both sides. Thereafter, the density remained approximately at the sham level.

### 2.4. No Changes in the Expression of the 5-HT Fibers in the Spinal Cord after TBI

The presence of 5-HT fibers in the lumbar gray matter in the 7-day TBI and sham rats was investigated. Some 5-HT fibers were present in the ventral horn, in the intermediate zone, and around the central canal on both sides in the TBI and sham rats. Qualitatively, no apparent distribution or density difference could be seen in the corresponding regions of the spinal cord on both sides in either the TBI or the sham rats (Figure 8A–D). To confirm this observation, we made a quantitative analysis of the 5-HT fiber densities in the ventral horn (Figure 8E). As shown in Figure 8F, there were no significant differences in the immunoreactive densities of the 5-HT fibers between the ipsilateral and the contralateral side of the spinal cord in either the TBI or the sham rats. The same was true when the same side was compared between the TBI and the sham rats. These results indicate that the limited focal brain injury in the sensorimotor cortex might not have affected the descending projection of the 5-HT fibers from the brain stem raphe nuclei to the spinal cord.

## 3. Discussion

The principal findings of this study were the changes in sustained hindlimb posture and walking pattern after a focal injury to the hindlimb sensorimotor cortex. The walking test showed a longer overall stride length in both hindlimbs after the brain injury, and the hind-paw placement angle was increased on both sides during walking. However, to our surprise, the sham surgery also induced an increased hind-paw placement angle on the contralateral side. All these motor deficits lasted at least 28 days, which was the longest time interval investigated in this study. The 5-HT2AR and 5-HT2CR expression showed no significant differences between the ipsilateral and contralateral side in either the TBI or the sham rats at any time point, indicating that 5-HT2AR and 5-HT2CR expression is probably not related to the development of HL-PA after brain injury. We did see variations in the 5-HT2AR and 2CR expression in the early phase, however. Thus, on both spinal sides, 5-HT2AR expression was significantly decreased on day 7 compared with the other time points, and 5-HT2CR expression was significantly decreased on day 3 compared with day 7. In the later phase, the expression levels recovered to approximate the levels in the sham rats. The absence of a clear trend in the expression of these receptors makes it difficult to speculate as to their role in the development of motor deficits.

The development of HL-PA after unilateral TBI was demonstrated in our previous studies through the flexion of the contralateral hindlimb [8,32]. However, although the rats survived up to 19 days after TBI, these studies did not focus on the time course of HL-PA development [8]. In the present study, the HL-PA changes over 4 weeks were investigated in five different time groups, with each group containing a nearly equal number of rats that were equally distributed between the TBI and sham subgroups. Despite some fluctuations, the magnitude of the HL-PA tended to increase with time, reaching a plateau by day 21 and then reducing slightly by day 28. An unexpected result was a decrease in the HL-PA magnitude at day 14, followed by an increase. This is difficult to explain from our data, but it may have been due to technical reasons, such as the different depths of anesthesia at different measuring points, or to biological reasons, such as the secondary damage (such as neuroinflammation and oxidative stress) that occurs at different times after a primary injury [33,34], and that leads to fluctuations in the recovery of motor function.

Previous studies with rat TBI models have used beam-walking and/or ladder-rung walking tests to examine the effects of brain injuries on balance control and limb coordination. In these studies, TBI rats were more likely to slip on the beam or rung bars than sham rats [35,36], indicating reduced control over balance and limb coordination. The walking test in the present study was performed on the ground, so the focus was on changes in walking pattern/gait rather than on balance. Gait changes, including stride length, stride speed and stride cadence, have long been used as parameters to evaluate the severity of motor impairments and the recovery of motor function after stroke and TBI [37,38,39,40,41]. Our walking test showed gait deficits after brain injury in the form of longer stride length and larger paw placement angle, indicating that brain injuries change walking patterns. Indeed, children with post-TBI also showed significantly greater variability in step time and step length compared with healthy controls [42], but based on studies with stroke patients, we would expect an asymmetric walking pattern and smaller stride length in the early phase after brain injury [43,44,45]. Although an asymmetric walking pattern was seen in the TBI rats in the current study (considering the increased hind-paw placement angle on the contralateral side), a longer stride length was seen in both hindlimbs and both forelimbs. The longer stride length was most likely induced by the contralateral hindlimb, and the increased stride length on the ipsilateral side and the forelimbs could have been due to interlimb coordination caused by propulsive force from the affected leg—a phenomenon called paretic propulsion symmetry [46,47]. In stroke patients, the increased step and stride length indicates the start of motor function recovery with training [48,49,50,51]. In our study, a significant stride length increase was already seen in the hindlimbs from 3 days after TBI. This may indicate that TBI and stroke have different recovery processes, or that the morphological and functional recovery in rodents is quicker than in humans due to rapid neuroplastic changes.

Our results showed a time-dependent increase in paw placement angle on both the contralateral and ipsilateral sides in the TBI rats. Foot placement abnormalities have been reported in patients with stroke, TBI, and spinal cord injury [52,53,54]. In animal models of TBI or corticospinal tract lesion, altered paw placement was also reported [36,55,56,57]. This suggests that TBI and stroke affect the coordination of the limbs and/or center of body mass, so that affected individuals need to adjust their limb or paw position laterally to maintain body balance during walking [58]. In our TBI model, only a small area was removed from the cortex, which would interrupt a small proportion of the corticospinal tract, but it still had relatively long-term detrimental effects on the pattern of movement. The corticospinal tract is not essential to generate basic locomotion rhythms in animals, but it is required for precise control over paw placement or limb trajectory [55]. One may expect that after unilateral TBI, the abnormal paw placement would only occur on the contralateral side; however, the same changes were seen on the ipsilateral side. Further investigation is needed to determine whether there were pathological changes in the spinal cord circuits that controlled the ipsilateral limbs, or whether this was just an adaptation to the limbs on the other side. A potential benefit of such a postural adjustment (the larger lateral angle) is that it might increase the stability of the body during walking.

An unexpected finding in this study was that the sham rats also showed a larger hind-paw placement angle on the contralateral side. This is difficult to explain, as there was no damage to the cortex in the sham rats. A possible explanation is that neurohormones were released in response to the injury to the tissues covering the brain. Recent evidence from us and our collaborators has shown that left-side brain injury caused the release of selected neurohormones (most likely including arginine-vasopressin and beta-endorphin), which caused hindlimb flexion on the right side in rats whose spinal cords were transected before TBI surgery [36]. It has also been demonstrated that the activation of opioid receptors can induce hindlimb flexion on different sides. For example, the activation of µ- and κ-opioids induced left hindlimb flexion in non-brain-injured rats, while the activation of δ-opioids induced right hindlimb flexion [32]. Opioids could be released under pain stimulation, such as when cutting the skin or removing bone from the skull. Thus, it is reasonable to speculate that pain stimulation under or after the operation might have released opioids from the hypothalamus/pituitary gland, and that this may have altered paw placement on the contralateral side in the sham rats.

One aim of the present study was to examine whether 5-HT2AR and 2CR would be upregulated after TBI, as seen after spinal cord injury [27,28,29]. However, we found no significant differences between the ipsi- and contralateral sides at any observed time points after TBI and sham surgery. Despite some fluctuations, there were also no significant differences between the TBI and sham rats when the same side of the spinal cord was compared. The small focal injury in the cortex presumably only had minor effects, if any, on the brainstem raphe nuclei that projected to the spinal cord. Indeed, when the 5-HT fiber immunoreactive density was analyzed in the TBI and sham rats, there were no significant differences between the sides or between the TBI and sham rats (Figure 8). The increased expression of 5-HT2 receptors is, to a large extent, related to the degree of the loss of 5-HT fibers in the spinal cord caused by injury. For example, complete spinal transection caused a high degree of 5-HT2AR upregulation in the spinal cord below the lesion both at mRNA and protein levels [28,59], while a cervical hemisection did not induce detectable changes in either 5-HT2AR or 2CR at the mRNA level [60]. In our TBI model, the 5-HT neurons in the brainstem and their projections to the spinal cord were not damaged, so the expression of 5-HT receptors should not be affected. We need to stress that immunohistochemistry is not the best technique for the quantitative analysis of the proteins in an organ. More sensitive techniques, such as radioimmunoassay, ELISA, or liquid-chromatography–tandem-mass-spectrometry, should be used to detect small changes in proteins. If the changes are significantly large, however, the differences can still be detected by analyzing the optical density of the immunoreactive products by using immunohistochemistry [27,28,29].

In conclusion, the current study showed that a focal brain injury to the hindlimb sensorimotor area caused motor deficits in posture and walking in rats. These deficits were manifested not only in the contralateral hindlimb but also in the ipsilateral hindlimb and forelimbs during walking, probably through a compensatory mechanism [3]. Although the injury area was small, the symptoms lasted at least up to 28 days, which was the longest time interval examined in this study. Our previous research indicated that it is spinal cord plasticity that underlies the development of motor deficits after brain injury, as the HL-PA remained in TBI rats after spinal transection [8]. The factors in the spinal cord underlying the motor deficits might be multiple. They may include intrinsic spinal neural circuit changes [17], primary afferent fiber reorganization [61], and/or asymmetric distributed opioid receptors in the spinal cord [36,62]. However, the expression of 5-HT2AR and 5-HT2CR is not likely to be related to the development of these motor deficits, as indicated by the results, although this does not rule out that plastic changes to other related monoamine transmitters and/or receptors may occur and play an essential role in the development of these motor deficits.

From this and our previous studies, we cannot speculate how long the motor deficits may last in t TBI rats subjected to this focal cortical lesion. At present, most research work using rodent TBI models, including ours, does not examine motor impairments past 2 months. Thus, in the future, it is necessary to assess the long-term effects on this model, over 6–12 months, for example [63]. In the present study, the limb movement impairments were assessed with a simple device—a home-made running tunnel on a piece of millimeter paper. Although this device is easy to produce, economical, and practical, it cannot be used for the high-fidelity analysis of limb movements. This would require more advanced tools capable of testing the temporospatial alteration of the movements, such as the multifactorial motor behavior assessment system to analyze limb kinematics, kinetics, and multi-directional forces, and electrophysiological metrics to decipher the motor problems in this model [64]. For gait analysis, automatic gait analysis devices such as DigiGait (MouseSpecifics, Inc.) and CatWalk XT (Noldus) can be used [65]. These devices analyze gait in different dimensions and, consequently, subtle motor deficits can be detected and analyzed. These in-depth analyses will no doubt provide valuable data for developing strategies for the treatment of motor deficits induced by TBI and stroke.

## 4. Materials and Methods

Adult male Sprague Dawley rats (Janvier Labs, Le Genest-Saint-Isle, France) weighing 250–600 g were used in the study. The animals received food and water ad libitum and were kept in a 12-hour day–night cycle at a constant environmental temperature of 21 °C (humidity 65%). The animals were randomly assigned to their respective experimental groups (see below) (Figure 9, Table 2). Approval for animal experiments was obtained from the Animal Experiments Inspectorate (permit # 2019-15-0201-0015). All animal experiments followed the guidelines of European Union Directive 2012/63/EU. 

### 4.1. TBI and Sham Surgery

The rats were given oral buprenorphine (Temgesic^®^ 485473, Indivior Europe, VA, USA) 0.4 mg/kg 1 h before the operation by mixing the drug with peanut butter; the drug acts over 24 h and was used for postoperative pain relief. The rats were anesthetized intraperitoneally (i.p.) with a mixture of ketamine 100 mg/kg (Ketaminol Vet 20 mg/mL, CAS: 6740-88-1, MSD Animal Health, Stockholm, Sweden) and xylazine 10 mg/kg (Rompun Vet 50 mg/mL, CAS: 7361-61-7, Elanco Denmark, Ballerup, Denmark) with a dose 0.25 mL/kg body weight. To avoid eye irritation (dry eyes) during surgery, 2 mg/g carbomer (ViscoTears, CAS: 9007-20-9, Bausch & Lomb Nordic AB, Stockholm, Sweden) was applied to eyes. Local anesthesia of 20 mg/mL lidocaine (Xylocain, CAS: 137-58-6, Aspen Nordic, Aspen, CO, USA) was given in the ears and subcutaneously on the surgical area.

The brain injury and sham operation were performed as described previously [8]. Briefly, the rats’ heads were fixated on a stereotaxic head holder. Under local anesthesia with lidocaine, the scalps were incised along the midline, opened, and stretched to expose the right side of the cranium. Next, a section of the cranium was marked with the coordinates 0.5–4.0 mm posterior to the bregma and 1.8–3.8 mm lateral to the midline. The marked cranial area was opened by drilling. The part of the cerebral cortex that is located below this opening includes the hind-limb representation area of the sensorimotor cortex, which was aspirated with a glass pipette (tip diameter ~0.5 mm) connected to an electrical suction machine (Craft Duo-Vec Suction unit, Rocket Medical Plc, Watford, UK). Care was taken not to suck off the white matter below the cortex. After the suction was complete, the bleeding was stopped using Spongostone (Lagaay Medical, Rotterdam, The Netherlands). The wound was closed using 3-0 vicryl suture (Ethicon, Raritan, NJ, USA), and lidocaine was applied on the operation area. The rats were each placed in a recovery cage for 24 h after the operation and were later returned to their home cage. For the sham operation, the same anesthetic and operative procedure was used, but the dura was kept intact, and the cortex was not ablated.

### 4.2. Postural Analysis

#### 4.2.1. Analysis of HL-PA

HL-PA was measured in all rat groups before euthanization. The magnitude of HL-PA was assessed as described previously [8]. HL-PA assessments were initiated by anaesthetizing the rats with 3.0% isoflurane (Attane Vet., CAS: 26675-46-7, Scanvet A/S, Fredensborg, Danmark) in a mixture of 0.3 L/min O_2_ and 0.6 L/min N_2_O gas. Each rat was placed in a prone position on millimeter paper. When the rat lost overall muscle tone, both hindlimbs were gently pulled for 5–10 mm to reach the same level and then set free. HL-PA magnitude was measured as a millimetric difference between the same digits of the two hind paws. This assessment was repeated five times for each rat, and the averaged value was taken as the magnitude of the rat’s HL-PA. Hindlimbs with shorter projections were regarded as flexed. Projection differences between the two hindlimbs over 2 mm were regarded as symmetrical [8].

#### 4.2.2. Analysis of Gait Pattern during Walking

The walking tests were only performed on the TBI and sham rats in the 28-day groups. The tests were performed before the operations (0 day) and 3, 7, 14, 21, and 28 days after the operations. As illustrated in Figure 10, the four paws of each rat were dipped in black ink on a stamp pad. Next, the rat was set to walk through a plexiglass tunnel that led into a dark box with a piece of millimeter paper in the bottom. The tunnel was custom-made out of 1-centimeter-thick transparent plexiglass and was 50 cm long, 10 cm high, and 10 cm wide [66]. Before running the real test, a training session was held to allow the rat to become familiar with the environment. This session consisted of the rat walking at least three times through the tunnel. For the real test, the procedure was repeated three times per rat per time point. After the test was completed, all pieces of the millimeter paper were scanned and analyzed using ImageJ program. Three different measurements were performed in this test: stride length of both the forelimbs and the hindlimbs (Figure 10C) and the placement angle of the hind paws (Figure 10D). Stride length was measured from the same point of a given paw during a stride. To measure paw placement angle, a line was drawn through the middle of the heel and extended through the middle digit, and the angle between this line and the straight sagittal line on the millimeter paper was measured.

### 4.3. Tissue Preparation

The rats were euthanized by perfusion fixation at 3, 7, 14, 21, and 28 days after surgery. When HL-PA and gait pattern had been assessed, each rat was administered 60 mg/100 g body weight sodium pentobarbital (Exagon Vet., Salfarm, CAS: 76-74-4, Kolding, Denmark), and an intracardial perfusion fixation was followed. The perfusion was started with 0.1 M phosphate buffered saline (PBS) followed by 100 mL/100 g body-weight fixative consisting of 4% paraformaldehyde (CAS: 30525-89-4, Sigma-Aldrich, Soeborg, Denmark) in 0.1 M PBS. Finally, the brain and entire spinal cord were dissected out and post-fixated in 4% paraformaldehyde for 24 h at 4 °C. The fixated brain and spinal cord tissues were then cryoprotected in a protective solution consisting of 30% sucrose (CAS: 57-50-1, Sigma-Aldrich, Soeborg, Denmark) diluted in PBS and 0.1% NaN_3_ for 24–48 h at 4 °C. The left ventral funiculus of the lumbar spinal cord was marked by a small cut with a blade to differentiate between the ipsilateral and contralateral side of the spinal cord and the brain-surgery side. The lumbar spinal cord (L1-L6) was cut transversely into 40-micrometer sections with a freezing microtome (Microm 34, Thermo Fisher Scientific, Roskilde, Denmark), cryoprotected in 30% sucrose with 0.1% NaN_3_ overnight, and subsequently stored at −80 °C. Every 24th section was stored together, aiming to include sections from all six segments of the lumbar spinal cords.

### 4.4. Histological Processing of the Brain Injury Sites

By visual inspection, the lesion sites in all the TBI rats were located on the right side; there was no cortical damage in any of the sham rats. To examine the actual lesion size of the brain injury in the cortex, five brains from TBI rats in different time groups (1 from 3 days, 1 from 7 days, 1 from 21 days, and 2 from 28 days) were cut transversely into 40-micrometer sections. Every fifth section across the lesion site was stained with 1% toluidine blue (Sigma-Aldrich, St. Louis, MO, USA,). The lesion volume was calculated from the sections containing the lesion site. From the stained sections, it was confirmed that the lesion did not affect the white matter below the cortex (Figure 11). The lesion area extended for 3.6–4.0 mm rostrocaudally and 2.4–3.0 mm mediolaterally. The lesions were 1.3–1.7 mm in depth. The lesion volumes were 6.2 ± 0.8 ${\mathrm{mm}}^{3}$ (mean ± SD; *n* = 5) after accounting for tissue shrinkage due to fixation (about 10%), which was similar to the values we reported in a previous study with the same animal model [68].

### 4.5. Immunohistochemistry

To investigate whether the focal brain injury altered 5-HT2AR and 5-HT2CR expression in the spinal cords, sections from the lumbar segments were processed for 5-HT2AR or 5-HT2CR immunohistochemistry from almost all the TBI and sham-operated rats (Table 2 shows the number of rats). Lumbar spinal cord sections from three pairs of 7-day rats (3 TBI and 3 sham rats) were also processed for 5-HT immunohistochemistry. The sections were processed using both avidin-biotin complex (ABC)-peroxidase and immunofluorescence [32,36].

#### 4.5.1. 5-HT2AR

One set of sections was processed for 5-HT2AR immunohistochemistry using the ABC-peroxidase method. To this end, every 24th section was incubated in PBS containing 0.3% H_2_O_2_ for 30 min and subsequently washed in PBS containing 0.1% Triton X-100 (PBS-T). To block non-specific bindings, a blocking serum containing 2% bovine serum albumin (BSA) and 5% normal goat serum (NGS) was added for 1 h. The sections were incubated in rabbit anti-5-HT2AR primary antibody (1:500; cat#: 24288, ImmunoStar Inc., Hudson, WI, USA) in the blocking solution for 48 h at 4 °C, and subsequently incubated in goat anti-rabbit biotinylated immunoglobulins (1:500; cat#: E0432, Dako, Glostrup, Denamrk) in PBS-T containing 1% BSA and 2% NGS for 1 h at room temperature (RT). The sections were further incubated in biotinylated tyramide (gift from Prof. Morten Møller, University of Copenhagen) diluted 1:500 in PBS and 0.005% H_2_O_2_ for 6 min to amplify the signal. Next, the sections were incubated in ABC (1:100; Vectastain Elite ABC kit, peroxidase (standard), Vector Laboratories, Burlingame, CA, USA) in PBS-T for 1 h at RT. The final reaction was performed by incubating the sections in 0.05 M tris buffer (pH 7.5) containing 0.05% diaminobenzidine and 0.01% H_2_O_2_ for 15 min. The sections were washed in tris buffer and mounted onto the microscope slides in 0.5% gelatin. After drying, the sections were dehydrated in 70%, 96%, and 99.9% alcohol and, subsequently, in xylene, before being coverslipped with DPX mounting medium (cat#: HX60964379, Merck Millipore, Burlington, MA, USA).

A further set of sections was processed for 5-HT2AR and ChAT double-fluorescence immunostaining. The purpose of the double-staining with ChAT was to locate the somatic and dendritic regions of the motoneurons in the spinal cords for further visualization of 5-HT2AR immunoreactivity. The sections were bleached before immunostaining with TrueView Vector (cat#: SP-8400, Vector Laboratories, Burlingame, CA, USA) for 5 min, according to the manufacturer’s instructions, to block autofluorescence. After washing in PBS-T, the sections were incubated in a blocking solution with 2% BSA and 5% normal donkey serum for 1 h, and then incubated in rabbit anti-5-HT2AR (1:200) and goat anti-ChAT (1:100; cat#: AB144P, Millipore, Burlington, MA, USA) primary antibodies in the blocking solution for 48 h at 4 °C. After washing, the sections were incubated in biotinylated donkey anti-rabbit IgG Alexa 594 (1:1000; cat#: A11058, Invitrogen, Waltham, MA, USA) and donkey anti-goat IgG Alexa 488 (1:1000; cat#: A11055 Invitrogen, Waltham, MA, USA) in the blocking solution for 1 h at RT. Next, the sections were washed with PBS-T and mounted onto the microscope slides in milli-Q water containing a small amount of PBS. After drying, the sections were coverslipped using Dako Fluorescence Mounting Medium (cat#: S3023, Dako, Glostrup, Denamrk).

#### 4.5.2. 5-HT2CR

Only double fluorescence was used for 5-HT2CR immunohistochemistry. The staining process was similar to that used for 5-HT2AR, apart from the use of different primary and secondary antibodies and different concentrations of antibodies and reagents. The sections were incubated in PBS containing 0.3% H_2_O_2_ for 30 min and then blocked in TBS-T with 2% BSA and 10% fetal bovine serum for 1 h. They were then incubated in rabbit anti-5-HT2CR (1:5000; cat#: ab32172, Abcam, Cambridge, UK) and goat anti-ChAT (1:100) in the blocking solution for 48 h at 4 °C. Next, the sections were incubated in biotinylated swine anti-rabbit (1:500; cat#: E0353, Dako, Glostrup, Denamrk) and ABC (1:100) for 1 h each in the blocking solution at RT and then incubated in biotinylated tyramide (1:10000; cat #: 6241, Tocris, Bristol, UK) in PBS and 0.005% H_2_O_2_ for 6 min, followed by 1 h of incubation each in donkey anti-goat IgG Alexa 488 (1:200; cat#: A11055, Invitrogen, Waltham, MA, USA) and streptavidin-conjugated Alexa 594 fluorophore (1:600; cat#: A11058, Invitrogen, Waltham, MA, USA) in the blocking solution at RT. The sections were thoroughly washed with PBS-T between the different steps. Finally, the sections were mounted onto the microscope slides and coverslipped.

#### 4.5.3. 5-HT

To investigate whether the brain injury affected the descending 5-HT innervation from the brainstem to the spinal cord, lumbar spinal sections from six 7-day rats (three TBI and three sham) were processed with fluorescence immunohistochemistry. After blocking in PBS-T with 10% fetal bovine serum and 2% BSA, the sections were sequentially incubated in primary antibody goat anti-5-HT (1:500; cat#: 20079, Immunostar, Hudson, WI, USA) for 48 h at 4 °C and in secondary antibody donkey anti-goat Alexa 594 (1:500; cat#: A11058, Invitrogen, Waltham, MA, USA) for 1 h at RT in the same solution. The sections were thoroughly washed with PBS-T between the different steps. Finally, the sections were mounted onto the microscope slides and coverslipped.

### 4.6. Data Analysis

The spinal sections were observed with a Leica DM6000B microscope (Leica Microsystems, Wetzlar, Germany). Color and black–white images were captured digitally (Leica DFC420 C Digital Camera System, Wetzlar, Germany). For quantitative analysis of the optical density of 5-HT2AR, 5-HT2CR, and 5-HT immunoreactivity, 8-bit black–white images were obtained with a grayscale density ranging from 0 to 255. The acquisition parameter settings of the microscope (magnification, exposure time, saturation, contrast, and gain) were identical for all sections from the TBI and sham rats in each animal group. No further manipulation (adjustment of brightness, contrast, etc.) was performed on the images prior to quantitative analysis. However, the pictures used in figures were processed with Photoshop to obtain a better quality for visual effects.

As described previously [29], the optical density of 5-HT2AR and 2CR was analyzed on black–white images using ImageJ software (version 1.53j, National Institutes of Health, Bethesta, MD, USA). We only analyzed the optical density in the ventral horns in which the motoneurons were contained. The images were imported into ImageJ, and the optical density in a 1200 × 1100-micrometr area in the ventral horn was measured automatically. For the analysis of 5-HT fiber immunoreactivity, the images were first thresholded so that only the pixels above the threshold level were counted as positive labeling elements [27]. The pixel area occupied by the 5-HT-positive elements was measured, and the percentage fractions were calculated. For each rat, 5–10 sections were analyzed, and the data were averaged and compared between the left and right side of the section and between TBI and sham rats on the same side in each time group.

Statistical analysis was performed using GraphPad Prism software (version 9, GraphPad Software, San Diego, CA, USA). For analysis of HL-PA, the mean values were compared between the TBI and the sham rats in different time groups with an ordinary one-way ANOVA and two-way ANOVA. The mean values of forelimb stride length, hindlimb stride length, and hind-paw placement angle were analyzed using a random intercept mixed model of linear regression. To analyze the difference in optical density between 5-HT2AR and 2CR immunoreactivity, we used paired t-test, linear regression, ordinary one-way ANOVA, and two-way ANOVA to compare the ipsilateral and contralateral sides of the spinal cords and to compare the TBI and sham rats. Ordinary two-way ANOVA was used to compare the averaged area of fraction for 5-HT fibers between TBI and sham rats and between two sides in each group. The group-averaged value was expressed as mean ± standard deviation (SD). The statistically significant level was set at *p* < 0.05.

## Figures and Tables

**Figure 1 ijms-23-05358-f001:**
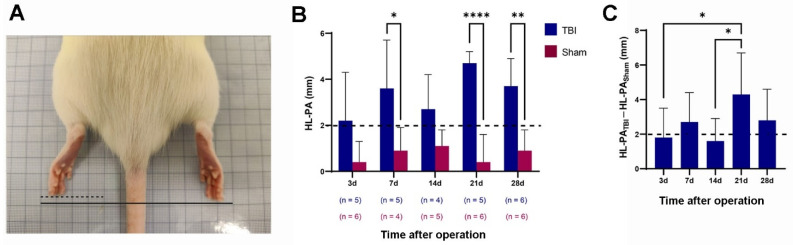
Traumatic brain injury (TBI)-induced formation of hindlimb postural asymmetry (HL-PA) and its retention over 4 weeks. (**A**) HL-PA was measured in millimeters as the difference between the projection points of corresponding digits on the two hindlimbs. (**B**) HL-PA after TBI and sham surgery in different animal groups. Horizontal dashed line indicates the 2-millimeter threshold for HL-PA. (**C**) HL-PA changes over 4 weeks after TBI (HL-PA_TBI_–HL-PA_sham_) in different animal groups. * *p* ≤ 0.05, ** *p* ≤ 0.01, **** *p* ≤ 0.0001 (B: two-way ANOVA; C: one-way ANOVA).

**Figure 2 ijms-23-05358-f002:**
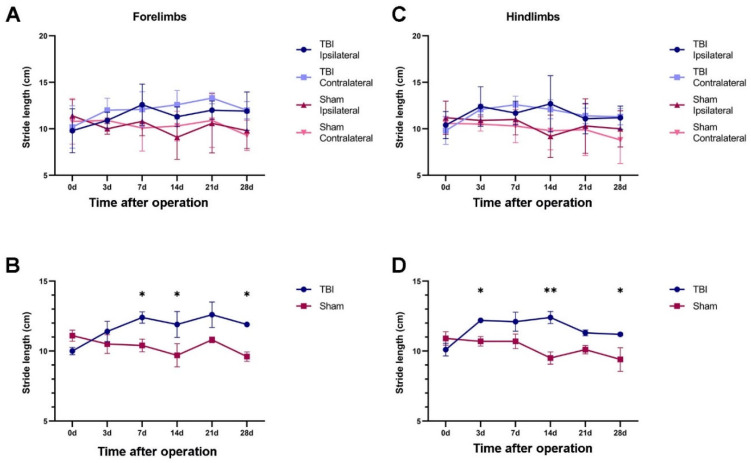
TBI-induced changes in stride length (SL) over 4 weeks. (**A**) The SLs of the forelimbs from TBI (*n* = 6) and sham rats (*n* = 6). (**B**) The SLs of the forelimbs with the values averaged from ipsilateral and contralateral sides showing the differences between the TBI and sham rats. (**C**,**D**) The same format as (**A**,**B**) showing the SL differences between the hindlimbs of the TBI and sham rats. * *p* ≤ 0.05, ** *p* ≤ 0.01 (two-way ANOVA).

**Figure 3 ijms-23-05358-f003:**
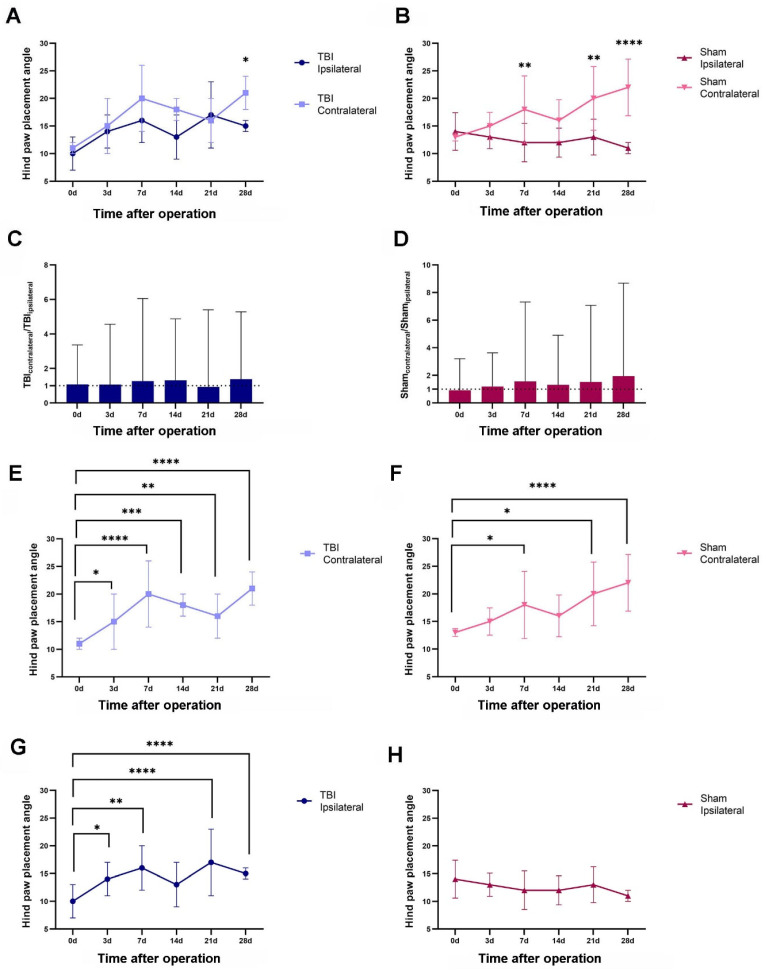
TBI and sham surgery-induced changes in hind-paw placement angle over 4 weeks. (**A**,**B**) The hind-paw placement angle measured from both ipsi- and contralateral sides after TBI (**A**) and sham surgery (**B**) at different time points (*n* = 6 each group). (**C**,**D**) Fold differences in hind-paw placement angle on the contralateral side relative to the ipsilateral side in TBI (**C**) and sham (**D**) rats. (**E**–**H**) The development course of the hind-paw placement angle after TBI surgery on the contralateral (**E**) and ipsilateral side (**G**) and after sham surgery on the contralateral (**F**) and ipsilateral side (**H**). * *p* ≤ 0.05, ** *p* ≤ 0.01, *** *p* ≤ 0.001, **** *p* ≤ 0.0001 (two-way ANOVA).

**Figure 4 ijms-23-05358-f004:**
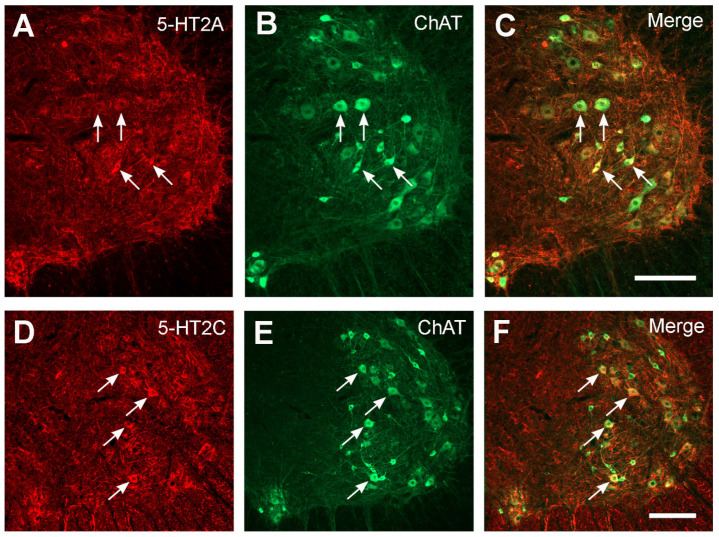
Double-fluorescence immunostaining showing 5-HT2A/2CR labeling in the lumbar spinal cord ventral horn motoneurons. (**A**–**C**) 5-HT2AR and ChAT fluorescence immunostaining in the ventral horn from a 7-day sham rat. (**D**–**F**) 5-HT2CR and ChAT fluorescence immunostaining in the ventral horn from a 14-day sham rat. Arrows point to 5-HT2AR/2CR and ChAT double-labeled motoneurons. Dorsal towards up and lateral to the right. Scale bar in (**C**), valid for (**A**–**C**), in (**F**), valid for (**D**–**F**), 200 µm.

**Figure 5 ijms-23-05358-f005:**
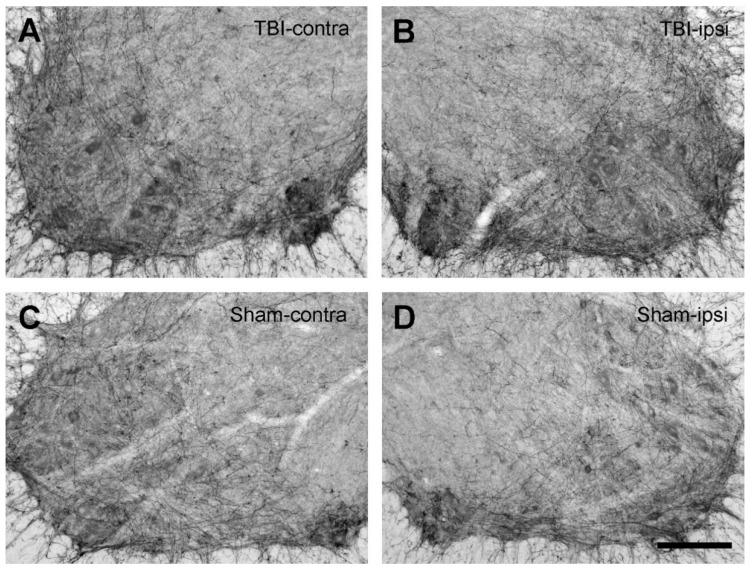
Black–white images from 5-HT2AR-immunostained sections with diaminobenzidine as a chromogen showing the labeling pattern in the ventral horns of the lumbar spinal cords (L5) in 28-day TBI and sham rats. (**A**,**B**) From the same spinal cord section showing the immunolabeling on the contra- and ipsilateral (right) side from a TBI rat. (**C**,**D**) From the same spinal cord section showing the immunolabeling on the contra- and ipsilateral (right) side from a sham rat. The ventral horn motoneuron regions were used for optical density analysis. Dorsal side upward and lateral either to the right (**A**,**C**) or left (**B**,**D**). Scale bar in (**D**), valid for all panels, 200 µm.

**Figure 6 ijms-23-05358-f006:**
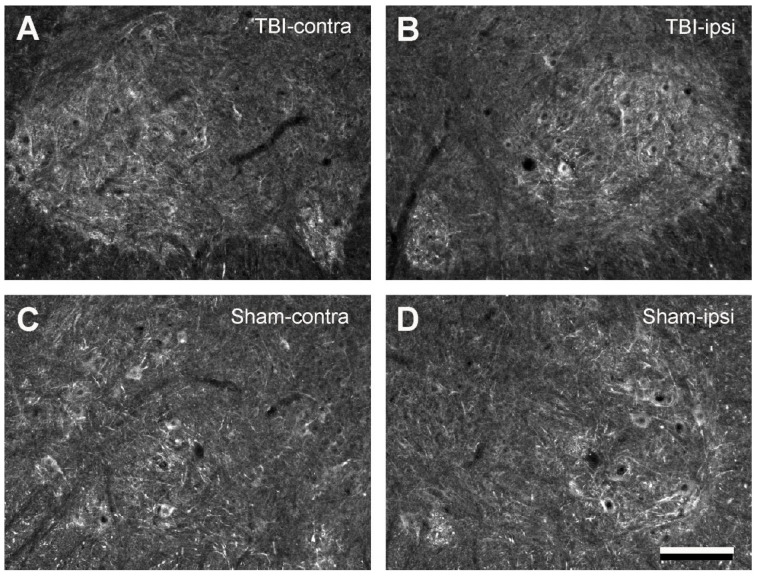
Black–white images from 5-HT2CR immunostained sections showing the labeling pattern in the ventral horns of the lumbar spinal cords (L5) in 14-day TBI and sham rats. (**A**,**B**) From the same spinal cord section showing the immunolabeling on the contra- and ipsilateral (right) side from a TBI rat. (**C**,**D**) From the same spinal cord section showing the immunolabeling on the contra- and ipsilateral (right) side from a sham rat. The ventral horn motoneuron regions were used for optical density analysis. Dorsal side upward and lateral either to the right (**A**,**C**) or left (**B**,**D**). Scale bar in (**D**), valid for all panels, 200 µm.

**Figure 7 ijms-23-05358-f007:**
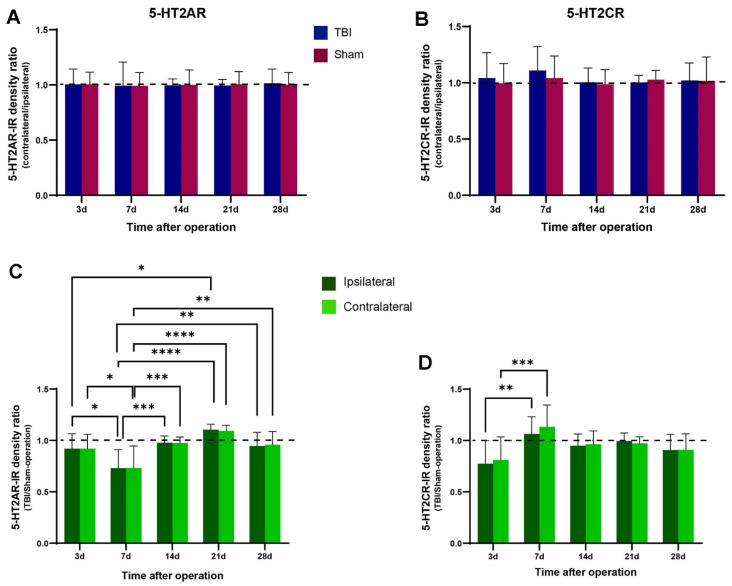
Quantitative analyses of 5-HT2AR and 2CR expression after TBI and sham surgery. (**A**,**B**) Histogram showing the quantitative data of 5-HT2AR immunoreactivity (IR) (**A**) and 5-HT2CR-IR (**B**) densities on the contralateral side relative to the ipsilateral side (normalized density value ± SD) in both the TBI and sham rats at different time intervals. There were no statistically significant differences between the ipsi- and contralateral sides in either TBI or sham groups at any time intervals. (**C**,**D**) Quantitative data showing 5-HT2AR-IR (**C**) and 5-HT2CR-IR (**D**) densities on the ipsi- and contralateral sides in TBI rats relative to the sham rats at different time intervals. There was a significant 5-HT2AR-IR density decrease on both sides of the spinal cord at 7 days compared with 3 days, after which the expression recovered to almost the same level as in the sham rats and stayed at this level until 28 days, albeit with some fluctuations. 5-HT2CR showed a significant density increase from 3 days to 7 days on both ipsi- and contralateral sides. * *p* ≤ 0.05, ** *p* ≤ 0.01, *** *p* ≤ 0.001, **** *p* ≤ 0.001 (two-way ANOVA).

**Figure 8 ijms-23-05358-f008:**
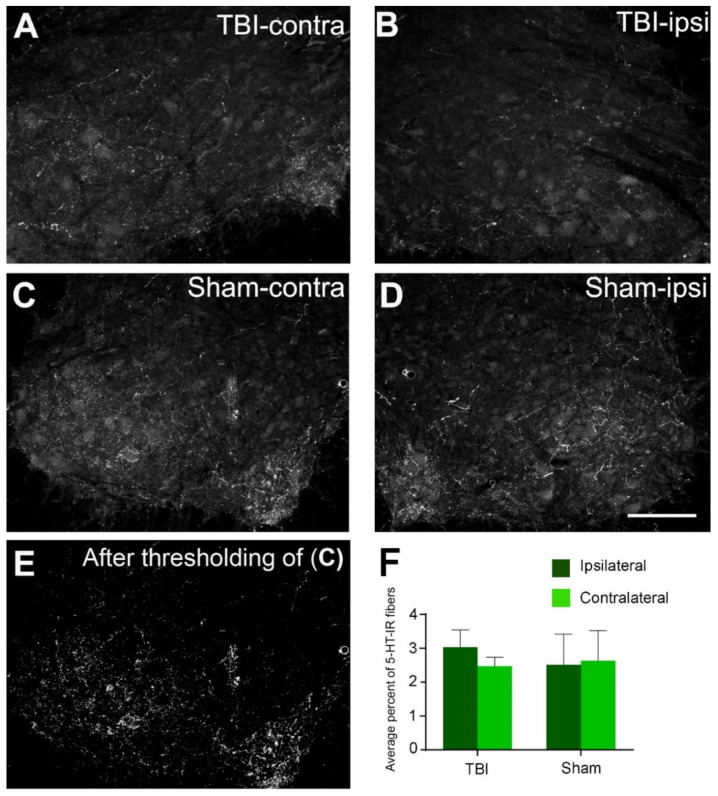
5-HT fiber immunoreactivity in the contralateral and ipsilateral ventral horns of the lumbar spinal ventral horns in 7-day TBI and sham-operated rats. (**A**,**B**) From the same spinal cord section showing 5-HT immunolabeling on the contra- (**left**) and ipsilateral (**right**) sides from a TBI rat. (**C**,**D**) From the same spinal cord section showing the immunolabeling on the contra- (**left**) and ipsilateral (**right**) sides from a sham rat. (**E**) Same image as in (**C**) after thresholding used for 5-HT immunoreactive density analysis. (**F**) There were no significant differences in 5-HT fiber areas between contra- and ipsilateral sides in either TBI or sham rats (two-way ANOVA). Dorsal side is upward for (**A**–**E**), lateral side is to the left for (**A**,**C**,**E**), and lateral side is to the right for (**B**,**D**). Scale bar in (**D**), valid for (**A**–**E**), 200 µm.

**Figure 9 ijms-23-05358-f009:**
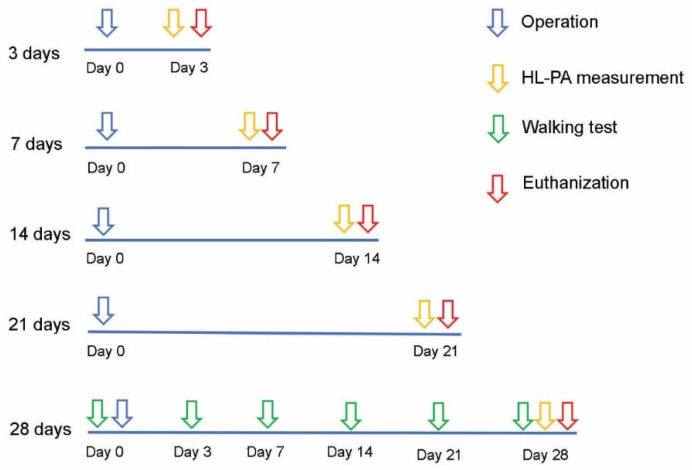
Illustration of the experimental setup and procedures.

**Figure 10 ijms-23-05358-f010:**
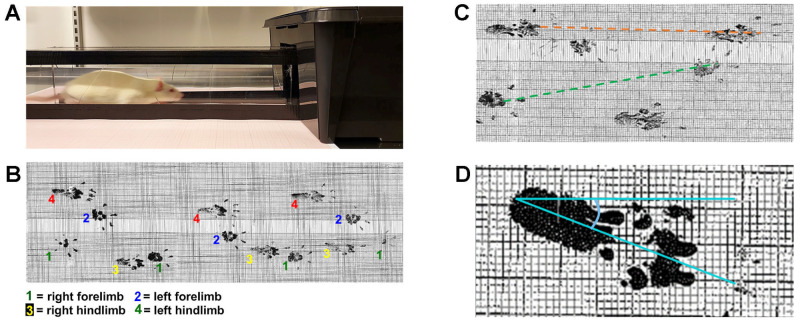
Walking test to detect gait pattern changes. (**A**) Setup of the device for walking test consisting of a running tunnel on a piece of millimeter paper and a dark box in the end of the tunnel. (**B**) Illustration of footprints of four paws from a 21-day TBI rat on a piece of millimeter paper. (**C**) Illustration of the stride length measurements for left forelimb (green) and left hindlimb [67] from a 3-day TBI rat. (**D**) Illustration showing how the hind-paw placement angles were measured. The straight line is parallel to the longitudinal axis of the millimeter paper, and the oblique line crosses the long axis of the paw (the midline from the heel to the middle digit).

**Figure 11 ijms-23-05358-f011:**
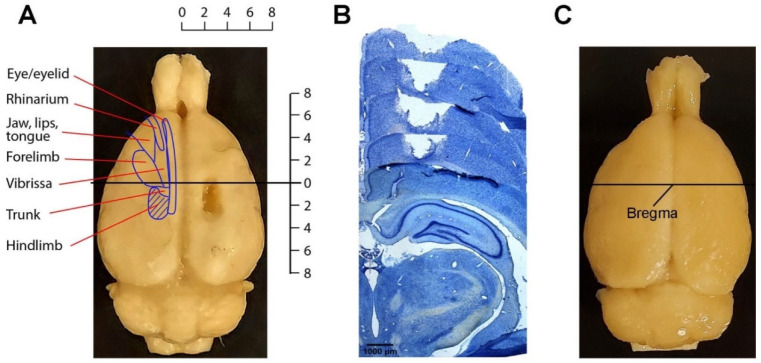
Lesion area in the hindlimb sensorimotor cortex. (**A**) Macrograph showing the lesion site in the cortex of the right hemisphere from a 28-day TBI rat. Delineation on left cortex represents somatotopically organized primary motor cortex (modified from [69]). The coordinates are in mm. The horizontal line crossing the zero point indicates the bregma level. (**B**) Five consequent toluidine blue-stained cortical sections at an equal distance (200 µm) across part of the lesion site from a 21-day TBI rat. (**C**) Macrograph showing a brain from a 28-day sham rat with no damage to the cortex.

**Table 1 ijms-23-05358-t001:** Stride length of forelimbs and hindlimbs in different animal time groups.

Limbs	Group	TBI		Sham	
		Contra	Ipsi	Contra	Ipsi
Forelimbs	3 days	12.0 ± 1.3	10.9 ± 0.9	10.9 ± 1.0	10.0 ± 0.6
	7 days	12.1 ± 1.9	12.6 ± 2.2	10.1 ± 2.5	10.8 ± 1.5
	14 days	12.6 ± 1.5	11.3 ± 1.1	10.3 ± 1.6	9.1 ± 2.4
	21 days	13.3 ± 0.6	12.0 ± 1.0	10.9 ± 2.9	10.6 ± 3.2
	28 days	12.0 ± 0.9	11.9 ± 2.1	9.3 ± 1.6	9.8 ± 1.9
Hindlimbs	3 days	12.1 ± 0.5	12.4 ± 2.1	10.5 ± 0.7	10.9 ± 0.5
	7 days	12.6 ± 0.9	11.7 ± 1.3	10.3 ± 1.8	11.0 ± 1.6
	14 days	12.1 ± 1.0	12.7 ± 3.0	9.8 ± 2.1	9.2 ± 2.3
	21 days	11.4 ± 0.8	11.1 ± 1.6	9.9 ± 2.8	10.3 ± 2.9
	28 days	11.3 ± 0.9	11.2 ± 1.2	8.8 ± 2.5	10.0 ± 1.9

Contra: Side contralateral to the injury; ipsi: side ipsilateral to the injury. TBI: traumatic brain injury. The unit is cm.

**Table 2 ijms-23-05358-t002:** The number of rats used in different groups and for different experimental procedures.

Survival Time	Total No. of Rats		HL-PA Analysis		Gait Pattern Analysis		5-HT2AR IHC		5-HT2CR IHC		5-HT IHC	
	TBI	Sham	TBI	Sham	TBI	Sham	TBI	Sham	TBI	Sham	TBI	Sham
3 days	6	6	5	6	-	-	6	6	6	6	-	-
7 days	9	9	5	4	-	-	6	6	5	5	3	3
14 days	6	6	4	5	-	-	6	6	6	4	-	-
21 days	6	6	5	6	-	-	5	5	5	5	-	-
28 days	6	6	6	6	6	6	6	6	6	6	-	-

HL-PA: hindlimb postural asymmetry, IHC: immunohistochemistry, TBI: traumatic brain injury.

## Data Availability

The data presented in this study are available on request from the corresponding author.

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
