# Peer review of "The Development of Hindlimb Postural Asymmetry Induced by Focal Traumatic Brain Injury Is Not Related to Serotonin 2A/C Receptor Expression in the Spinal Cord"

_ijms, 2022, doi:10.3390/ijms23105358_

Round 1

Reviewer 1 Report

The research article entitled " Development of Hindlimb Postural Asymmetry Induced by Focal Traumatic Brain Injury Is Not Related to Serotonin 2A/C Receptor Expression in the Spinal Cord”, by Andersen et. al. researched the development of limb postural changes and their relationship to expression of serotonin (5-HT) 2A and 2C receptors in the spinal cord after focal traumatic brain injury. In my opinion, the article is well written, the introduction provides sufficient detailed knowledge regarding the pathological mechanisms underlying these motor deficits and its relationship to expression of serotonin (5-HT) 2A and 2C receptors in the spinal cord. In conclusion, the current study showed that a focal brain injury to the hindlimb sensorimotor area caused motor deficits in posture and walking in rats. The drawback of the study is that authors used a homemade device to study, the limb movement impairments but which cannot be used for high- fidelity analysis of limb movements, as acknowledged by the authors as well. The article is well presented and discussed with elaborated figures to support the results. I trust that the manuscript will be of interest to potential readers.

Author Response

Thank the reviewer very much for the nice review for our manuscript. We acknowledged that the homemade device for walk test is not ideal. This has been pointed out in the Discussion. This is due to the current experimental condition in the lab. In the future experiments we will try to use high-fidelity analysis of limb movements if the economic situation allows, or through collaboration with other labs where such facility is accessible. 

Reviewer 2 Report

In this paper Andersen et.al. show that serotonin 2A/C receptor expression in the spinal cord has no role in hind limb postural asymmetry induced by TBI. 

Excellent work but here are some critical comments:

1) The injury site is absolutely not clear. How was the injury made? Mere mentioning suction and citation is not sufficient explanation.

2) The fluorescence microscopy is not of good especially Fig 4. It looks like false filter color instead of  fluorescence. Please fix it.

3) Did you correct for auto fluorescence? If, yes, please mention it. 

Look forward to read the revised version.

Author Response

Responses

“In this paper Andersen et.al. show that serotonin 2A/C receptor expression in the spinal cord has no role in hind limb postural asymmetry induced by TBI. 

Excellent work but here are some critical comments:”

Response: Thank you very much for judging our manuscript as “excellent work”. We are very appreciated.

  • “The injury site is absolutely not clear. How was the injury made? Mere mentioning suction and citation is not sufficient explanation.”

Response: This is a good point. The injury procedure was actually described in the text. Maybe it is not clear enough. We have now reformulated the drafting so that it reads more understandable (red text in page 16).

  • “The fluorescence microscopy is not of good especially Fig 4. It looks like false filter color instead of fluorescence. Please fix it.”

Response: This problem may be caused by the conversion of Word file to PDF file. Also, in the first-time submission we used jpg file instead of tiff file to place it in the Word file. The original file is in tiff, and we will submit the tiff file for the publication in the final stage. The original tiff file is in 300 dpi and the color looks fine. Now I have adjusted the color a bit using Photoshop and attached the figure in this letter (see the attached figure in the Word file). If you still see a poor color, I can send you a tiff file. (I also found that on different computers the color looks a bit different).

  • “Did you correct for auto fluorescence? If, yes, please mention it.”

Response: The answer is “no”. The reason is that we did not see auto fluorescence in our sections, maybe due to the rats used in our study were relatively young. A piece of evidence to verify this is that in the negative control (without primary antibody) staining we did not see any fluorescence labeling (see the pictures in the attached Word file). We have added this description in the image analysis (red text in page 20).

Reviewer 3 Report

In the present research paper, the Authors used an established unilateral TBI animal model in which the hindlimb representation area of the sensorimotor cortex was ablated. Although this TBI model has been shown to induce hindlimb postural asymmetry (HL-PA) on the contralateral side for varying periods after injury, a time course study of its development is lacking. Therefore, They further examined whether HL-PA changed over a period of 4 weeks and whether the injury affected the animal’s walking pattern over the same period. Finally, the Authors evaluated whether the expression of 5-HT2ARs and 5-HT2CRs in the lumbar spinal cord was affected, with the aim of evaluating their potential roles in the development of motor deficits after TBI.

Overall, I found the oresent study of an outstanding quality, timely, well conducted and scientifically sound: it might have significant implication in TBI research. 

I've read the paper many times and I cannot found any fault or suggestion aimed to improve the quality. Maybe my only question is why adult male Sprague Dawley rats were used and not other types?

Author Response

Thank the reviewer very much for the nice review. I will try to answer the only question raised by the reviewer: "Maybe my only question is why adult male Sprague Dawley rats were used and not other types?"

Reply: Since we started to use this brain injury rat model to do research we have always used male adult Sprague Dawley rats. With this model we have published 8 papers in different journals. To make the results comparable to other studies we have chosen the same rat strain, sex and age in the present study. It is definitely a good idea to test other strains to see whether different strains give different motor symptoms following a same kind of brain injury. Also it is extremely important to see whether in female rats the responses are the same with the male rats when suffering the same kind of brain trauma. All these are in our future experimental plan.

Round 2

Reviewer 2 Report

The authors have answered all the issues and made satisfactory changes to the manuscript. This paper is ready to be accepted.